# Epitope Spreading in Immune-Mediated Glomerulonephritis: The Expanding Target

**DOI:** 10.3390/ijms252011096

**Published:** 2024-10-16

**Authors:** Camillo Tancredi Strizzi, Martina Ambrogio, Francesca Zanoni, Bibiana Bonerba, Maria Elena Bracaccia, Giuseppe Grandaliano, Francesco Pesce

**Affiliations:** 1Department of Translational Medicine and Surgery, Università Cattolica del Sacro Cuore, 00168 Rome, Italy; martina.ambrogio01@icatt.it (M.A.); bibiana.bonerba@policlinicogemelli.it (B.B.); giuseppe.grandaliano@unicatt.it (G.G.); 2Nephrology, Dialysis and Transplantation Unit, Fondazione Policlinico Universitario A. Gemelli IRCCS, 00168 Rome, Italy; 3Department of Nephrology, Dialysis, and Kidney Transplantation, Fondazione IRCCS Ca’ Granda Ospedale Maggiore Policlinico, 20122 Milan, Italy; francesca.zanoni@policlinico.mi.it; 4Division of Renal Medicine, Ospedale Isola Tiberina-Gemelli Isola, 00186 Rome, Italy; mariaelena.bracaccia@fbf-isola.it

**Keywords:** epitope spreading, autoimmune glomerulonephritis, glomerular disease progression, epitope-specific immune response, targeted immunotherapy

## Abstract

Epitope spreading is a critical mechanism driving the progression of autoimmune glomerulonephritis. This phenomenon, where immune responses broaden from a single epitope to encompass additional targets, contributes to the complexity and severity of diseases such as membranous nephropathy (MN), lupus nephritis (LN), and ANCA-associated vasculitis (AAV). In MN, intramolecular spreading within the phospholipase A2 receptor correlates with a worse prognosis, while LN exemplifies both intra- and intermolecular spreading, exacerbating renal involvement. Similarly, ANCA reactivity in AAV highlights the destructive potential of epitope diversification. Understanding these immunological cascades reveals therapeutic opportunities—targeting early epitope spreading could curb disease progression. Despite promising insights, the clinical utility of epitope spreading as a prognostic tool remains debated. This review provides a complete overview of the current evidence, exploring the dual-edged nature of epitope spreading, the intricate immune mechanisms behind it, and its therapeutic implications. By elucidating these dynamics, we aim to pave the way for more precise, targeted interventions in autoimmune glomerular diseases.

## 1. Autoimmunity and Epitope Spreading

Autoimmunity manifests through precise targeting of specific sub-molecular structures within self-antigens known as epitopes. This results in their erroneous identification as foreign, leading to significant tissue damage and persistent inflammation. As the immune response progresses, it can expand from dominant to less dominant epitopes, a process known as epitope spreading [1]. This phenomenon can occur as a consequence of the diversification of the immune response that ensues following self-tissue damage in autoimmune reactions. The hierarchy of dominant and cryptic epitopes—antigenic regions that become more accessible during an inflammatory immune response—is the result of a combination of factors, including differential protein processing and antigen presentation by various antigen-presenting cells, the availability of epitope-specific T cells, and the mechanisms of central and peripheral tolerance [2,3]. The occurrence of tissue damage is contingent upon the presence of a trigger, which may be infection, organ transplantation, or autoimmunity (whether organ-specific or systemic). This damage, in conjunction with the ensuing inflammatory response, gives rise to a hierarchical cascade of autoreactive T-cell specificities, thereby enabling cryptic or sequestered epitopes to be processed and presented, contributing actively to the ongoing disease pathology [3]. Epitope spreading optimizes the immune response against foreign antigens, enhancing pathogen neutralization and clearance [3] and, indeed, it is essential for an effective adaptive immune response and for improving efficiency in tumor clearance; however, it also contributes to the development of autoimmune disorders, making it a double-edged sword [4].

### 1.1. The Mechanism of Epitope Spreading

The process of spreading is initiated upon antigen recognition by immune cells, with B cell activation driven by CD4+ T cells or specific antigen binding, and it ramps up through clonal expansion and affinity maturation, enabling the presentation of novel cryptic epitopes to CD4+ T cells via MHC class II molecules, thereby broadening the immune response over time [4]. Epitope spreading can occur in two distinct forms, namely, intramolecular and intermolecular [5]. Intramolecular spreading, which involves targeting different epitopes within the same molecule, is promoted by affinity maturation, whereby B cells with higher affinity for different epitopes are selected. Moreover, endocytic processing and MHC class II presentation facilitate this process by displaying previously unrecognized epitopes [6]. Conversely, intermolecular spreading enables the targeting of epitopes across multiple antigens, allowing T cells specific to one epitope to activate B cells targeting other antigens, thereby diversifying the antibody response against antigens that were not originally involved in either the B cell or the T cell response [7]. Furthermore, molecular mimicry cross-reactivity, which results in self-tolerance breakdowns when an autoantigen resembles a pathogen antigen closely enough that antibodies generated against the pathogen also target the autoantigen, can be followed by epitope spreading, thereby contributing to the development of autoimmune disease [4,8,9].

### 1.2. Epitope Spreading in Experimental Models

Epitope spreading has been observed in numerous autoimmune diseases, including systemic lupus erythematosus (SLE) [10,11], rheumatoid arthritis [12], type 1 diabetes [13], multiple sclerosis [14], Sjögren’s syndrome [15,16], Graves’ disease [17,18], scleroderma [19], dilated cardiomyopathy [20], pemphigus [21,22], bullous pemphigoid [23], and myasthenia gravis [24]. Several studies have provided compelling evidence to support the role of epitope spreading in autoimmune glomerulonephritis immuno-pathogenesis. In experimental models, Shah et al. demonstrated intramolecular spreading in Heymann nephritis. This was achieved through the immunization of subjects with the L6 fragment of megalin (gp330), a complex antigen with four discrete ligand-binding domains that may contain epitopes to which pathogenic autoantibodies are directed. This immunization initially resulted in the production of autoantibodies reactive with L6, recognizing, over time, additional epitopes on other ligand-binding domains of megalin with a correlation to increased proteinuria [25]. Similarly, Bolton et al. investigated epitope spreading in experimental autoimmune glomerulonephritis (EAG) induced by a T cell epitope of Goodpasture’s antigen. Their findings demonstrated that the immune response spread to other epitopes within the α3(IV)NC1 domain, contributing to disease progression [26]. Ross et al. investigated intermolecular spreading in a rat model of anti-GBM glomerulonephritis. Immunization with a T cell epitope of collagen 4α3 resulted in the production of autoantibodies that recognized multiple new epitopes on other GBM proteins, though the autoimmune inflammation remained confined to the glomeruli [27]. The findings are additionally supported by clinical evidence. In anti-GBM disease, autoantibodies initially target the NC1 domain of the α3 chain of type IV collagen and subsequently recognize additional epitopes on other chains, which correlates with increased severity and progression [28]. In both membranous nephropathy (MN) and SLE, initial immune responses to specific epitopes subsequently spread to additional epitopes, thereby exacerbating the disease process [25,28]. While current treatments rely on broad-spectrum immunosuppressants, targeted therapies addressing epitope spreading are still under research, and further studies are required to assess their safety and efficacy. A deeper understanding of these mechanisms could lead to more effective, targeted therapeutic strategies.

## 2. Membranous Nephropathy

Membranous nephropathy (MN) is defined by thickened glomerular capillary walls due to immune complex deposition on the outer basement membrane, primarily involving IgG and complement components, including the membrane attack complex. This immunological process disrupts podocyte structure, leading to significant proteinuria. The autoimmune nature of MN is elucidated by the recognition of a self-structure, the phospholipase A2 receptor 1 (PLA2R), as a pathogenic antigen [29]. Patient outcomes vary, with one-third achieving spontaneous remission, another third experiencing persistent proteinuria, and the final third progressing to advanced kidney failure [30]. MN has traditionally been classified as either primary or secondary based on the presence or absence of an identifiable underlying cause, such as systemic autoimmune disease, cancer, or infections [31]. The recent identification of numerous autoantigens associated with MN, including PLA2R, neural epidermal growth factor-like protein 1 (NELL1), exostosin (EXT1/2), proprotein convertase subtilisin/kexin type 6 (PCSK6), thrombospondin type-I domain-containing 7A (THSD7A), protocadherin 7 (PCDH7), semaphorin 3B (SEMA3B), neural cell-adhesion molecule 1 (NCAM1), and others [32], has blurred the lines between primary and secondary categorization. As a result, an antigen-based classification system is now preferred, recognizing that no single target antigen definitively distinguishes between primary and secondary forms of MN [33].

### 2.1. Evidence of Intramolecular Epitope Spreading in PLA2R-Associated MN

PLA2R is a type I transmembrane receptor belonging to the mannose receptor family. It is an N-glycosylated protein composed of an N-terminal cysteine-rich domain (CysR), a fibronectin type II domain (FNII), eight C-type lectin-like domains (CTLDs), a transmembrane domain (TM), and a short intracellular C-terminal tail (IC) [34]. It is capable of continuous endocytic recycling in clathrin-coated pits [35] through pH-dependent conformational changes necessary to allow ligand binding at physiological pH and cargo release at the more acidic pH of endosomes and lysosomes before returning to the cell surface, suggesting a possible role for PLA2R as a cargo protein, although the functional consequences in the podocyte remain speculative [36]. Beck et al. were the first to demonstrate the role of PLA2R as a primary target antigen in idiopathic MN [34]. Their findings revealed the presence of IgG antibodies, predominantly IgG4, that reacted with a reduction-sensitive epitope on PLA2R, similar to megalin in rat models [37] and neutral endopeptidase in alloimmune neonatal MN [38], acting as a target for antibody binding and complement-mediated podocyte injury or as receptor agonists or antagonists, altering podocyte architecture and barrier function, ultimately leading to proteinuria [34]. A decade ago, Kao et al. sparked interest in the role of epitope recognition in PLA2R-MN. Their study identified the immunodominant antigenic epitope responsible for autoantibody binding within the CysR-FnII-CTLD1 domain (1–3 construct) [39]. They demonstrated that the absence of this domain resulted in a lack of autoantibody recognition, supporting its role as a universal antigen domain. This finding led to the envisioning of developing CysR-FnII-CTLD1 domain ELISA assays for diagnosis and prognosis, comparable to full-length PLA2R. In the same year, Fresquet et al. further refined the understanding of autoantibody binding by focusing on the CysR domain. This resulted in the identification of a 31-amino-acid sequence within CysR that was capable of blocking the autoantibody binding site, potentially localizing the humoral epitope to this region [40]. This sequence was identified as a potential target for immunoadsorption columns, offering a promising avenue for removing anti-PLA2R antibodies, particularly in patients resistant to standard immunosuppressive therapies. Subsequently, Seitz-Polski et al. identified distinct epitopes within CysR, CTLD1, and CTLD7 that provoke reactivity against anti-PLA2R antibodies, introducing the concept of “intramolecular epitope spreading” in MN [41]. They highlighted CysR as the primary epitope, with spreading to CTLD1 and CTLD7. Anti-PLA2R1 reactivity against CysR was linked to favorable outcomes, while reactivity against CTLD1 and CTLD7 correlated with active disease, more proteinuria, and faster progression to ESRD. Temporal variability in epitope profiles showed anti-CTLD1 and anti-CTLD7 antibodies disappearing during remission and reappearing during relapse, whereas anti-CysR indicated stable disease. These findings suggest that anti-PLA2R1 antibodies initially target CysR in early MN, with spreading to CTLD1 and CTLD7 leading to a more active disease state [41]. Reinhard et al. corroborated the previously identified epitopes in the CysR, CTLD1, and CTLD7 domains and identified a novel epitope in the CTLD8 domain [42], demonstrating that the CysR and CTLD1 domains are preferentially recognized, which is likely due to their greater accessibility for antibody binding, though challenging the independent role of epitope spreading as a prognostic indicator.

### 2.2. Debating the Prognostic Significance of Intramolecular Spreading in PLA2R-MN

The early stages of PLA2R-MN (pauci-symptomatic) involve the initial targeting of the outermost epitope, CysR. As the disease progresses, antibodies develop against inner epitopes, such as CTDL1 through to CTDL8 [43], through intramolecular spreading, especially in the presence of a second immune challenge as allergy, infection [44], or exposure to organic solvents and asbestos, with a dose-dependent effect [45] (Figure 1). Nonetheless, the prognostic value of epitope spreading in MN remains highly debated, with the existing literature suggesting that the correlation between epitope spreading and worse prognosis is more closely related to overall PLA2R antibody titers [46,47,48], which often appear within the year before diagnosis [49]. It remains unclear whether this truly reflects epitope spreading or the concurrent targeting of multiple epitopes from the onset of the disease. The potential of antibodies specific to PLA2R epitope regions as superior predictors of disease outcomes compared to overall anti-PLA2R levels warrants further study. As this prognostic role could significantly impact treatment strategies, clarification is key. In the following sections, we dissect the existing literature with articles both in favor of and against the prognostic role of epitope spreading in PLA2R-MN.

#### 2.2.1. In Favor

Seitz-Polski et al. [50] utilized ELISAs to quantify epitope reactivity, categorizing patients enrolled in the GEMRITUX trial into four distinct groups: CysR, CysRC1, CysRC7, and CysRC1C7. At baseline, 65.5% of the patients exhibited epitope spreading beyond CysR, which was found to correlate with higher anti-PLA2R1 antibody titers. Patients with titers above 369.5 RU/mL (Euroimmun ELISA) were spreaders. After six months, lower CTLD7 reactivity and reduced epitope spreading were linked to higher remission rates, while PLA2R1-Ab titers were not significant. A decrease in both titers and spreading predicted remission at the final follow-up. In a different study [51], the authors confirmed that the absence of epitope spreading at baseline was associated with a higher rate of clinical remission, suggesting that patients with anti-PLA2R1 titers above 321 RU/mL should receive high-dose rituximab, as 95% of these patients are likely spreaders. These results indicate that patients exhibiting significant epitope spreading should be treated promptly and aggressively, with options such as rituximab, while those with low PLA2R1-Ab levels but substantial spreading should also be treated at diagnosis. Zhou et al. [52] employed the TRFIA assay to measure PLA2R domain-specific antibody titers, thereby demonstrating that PLA2R-CTLD1-IgG4 antibody levels were independently associated with proteinuria remission at six months, in contrast to PLA2R-IgG levels. It is noteworthy that one patient who exhibited negative baseline epitope reactivity and subsequently became an epitope spreader by six months continued to experience persistent nephrotic syndrome and treatment resistance despite low anti-PLA2R antibody titers [52], suggesting that domain-specific antibodies may better predict remission. Additionally, an individualized model was also developed to predict six-month proteinuria remission, identifying proteinuria and PLA2R-CTLD1-IgG4 as key factors, demonstrating superior predictive capacity in both training and validation cohorts, thereby demonstrating accuracy and practicality for clinical use [52].

#### 2.2.2. Against

All patients included in the population studied by Reinhard et al. exhibited at least two epitope regions targeted by PLA2R1 antibody at the time of renal biopsy, suggesting that either early epitope spreading or a multispecific immune response may be present from the onset of the disease [42]. In their study, the impact of epitope spreading on treatment response and prognosis was not observed, as the immune response had already targeted multiple epitopes at the time of diagnosis, and it was concluded that the most effective treatment would have to address all pathogenic PLA2R epitopes rather than just one domain-specific antibody. Domain-specific antibody levels did not offer greater prognostic value than total PLA2R1 antibody levels, which were predictive of clinical outcomes. To corroborate this finding, 31 patients exhibited spontaneous remission of proteinuria despite the presence of antibodies directed against both the N- and C-terminal PLA2R domains, with significantly lower baseline total PLA2R1-ab levels, highlighting the importance of total PLA2R1 antibody levels over multiple epitope recognition in predicting clinical outcomes. In addition, the study also examined whether specific PLA2R1 epitope antibody levels or recognition patterns could explain MN clinical variability. A PLA2R1 domain-specific ELISA showed that CysR and CTLD1 domains were preferentially recognized, consistent with serum dilution experiments that indicated that higher dilution reduced C-terminal domain recognition due to better N-terminal region accessibility for antibody binding. Liu et al. [53] revealed that the majority of IMN patients exhibited antibody reactivity to more than two epitopes. Following a six-month course of treatment, CysR-specific IgG antibodies were observed to be higher in the remission group in comparison to the non-remission group. Conversely, CTLD1/CTLD6-7-8-specific IgG antibodies demonstrated no statistically significant differences between the two groups. Ruggenenti et al. [48] conducted a longitudinal study on patients with PLA2R1-MN and persistent nephrotic syndrome who had not received immunosuppressive therapy for a minimum of six months. Their findings indicated that poorer outcomes were independently associated with higher baseline anti-PLA2R1 and anti-CysR antibody titers but not with anti-CTLD1, 7, and 8 titers or multidomain antibody recognition. Importantly, multidomain recognition did not correlate with proteinuria severity, hypoalbuminemia, hypoproteinemia, or dyslipidemia. Recognition of domains beyond CysR proved clinically irrelevant. Post-treatment, patients who achieved remission showed greater depletion of anti-CysR antibodies, a trend also seen with anti-CTLD1, 7, and 8 antibodies. Although higher anti-CysR levels were linked to more multidomain recognition, this did not significantly impact outcomes. The study also proposed that multidomain recognition might reflect early epitope spreading, enhancing immune response. However, the shorter proteinuria duration in multidomain recognizers challenges this view. The similar proportion of multidomain recognizers in patients treated with rituximab as first- or second-line therapy suggests early epitope spreading.

### 2.3. Therapeutic Implications

In light of these findings, epitope spreading and rituximab dosage are of paramount importance in the selection of treatment protocols for PLA2R1-related MN. While the KDIGO guidelines suggest a six-month wait before immunosuppressive treatment for patients with a CysR-restricted profile, this may be appropriate for those who are likely to achieve spontaneous remission [44]. However, for patients with highly active disease and epitope spreading, who are unlikely to remit spontaneously or respond to low-dose rituximab, this approach could reduce their likelihood of remission. The evaluation of the effect of rituximab on epitope profiles and spreading demonstrated a significant reduction in spreading along with a higher incidence of ‘reverse’ spreading from CysRC1 and/or CysRC7 to solely CysR or negative, in comparison to non-immunosuppressive anti-proteinuric treatment [50]. Rituximab dosing and infusion timing and their impact on remission outcomes in nephrotic syndrome were analyzed by comparing the NICE and GEMRITUX protocols. The NICE protocol involves two 1 g infusions of rituximab at 2-week intervals after 6 months of symptomatic antihypertensive and antiproteinuric treatment or earlier in the case of acute complications according to the KDIGO guidelines. The GEMRITUX protocol includes six months of symptomatic therapy, followed by two 375 mg/m^2^ Rituximab infusions a week apart if nephrotic syndrome persists, indicated by a Urinary Protein-to-Creatinine Ratio > 3.5 g/g and albumin < 30 g/L [54]. Results showed that higher cumulative doses (2 g in NICE vs. 1.4 g in GEMRITUX) and bi-weekly infusions led to more complete remissions by month 6. The absence of remission was linked to lower rituximab levels, higher CD19 counts, and elevated anti-PLA2R1 antibodies at month 3, highlighting the need to maintain higher drug levels for remission. The study suggested that the higher rituximab dose in the NICE protocol reduced epitope spreading by month 6, with potential reversal benefits, recommending high-dose rituximab for patients with anti-PLA2R1 titers above 321 RU/mL. An ongoing trial by Brglez et al. [44] (ClinicalTrials.gov ID NCT03804359) aimed to evaluate the effectiveness of personalized treatment in comparison to the GEMRITUX protocol for nephrotic MN patients by stratifying them according to their epitope profile and adjusting the timing and dosage of rituximab accordingly. The personalized treatment protocol tailors the approach based on specific epitope activity. For patients with restricted anti-CysR activity, the initial six months follow the same symptomatic treatment. If nephrotic syndrome continues, these patients also receive two rituximab infusions (375 mg/m^2^) one week apart. Patients with anti-CTLD 1/7 activity, either at the outset or after six months, receive two higher-dose rituximab infusions (1 g each) two weeks apart at the start and possibly again at the six-month mark. Preliminary results show [55], with no significant differences in age, gender, albumin levels, UPCR, anti-PLA2R1 titers, and the rate of PLA2R1 epitope spreading at baseline between the two groups; at 12 months, 34% of patients in the GEMRITUX group and 69% in the PMMN group significantly achieved partial clinical remission, defined as UPCR < 3.5 g/g with a decrease of more than 50% from baseline, improvement or normalization of serum albumin, and a serum creatinine increase of less than 20%. There was no difference in remission rates between the groups for patients with single-domain recognition. However, multi-domain recognizers (spreaders) were statistically more likely to achieve remission with the personalized protocol. The rate of complete clinical remission, defined as UPCR < 0.3 g/g and normal albumin, was close to statistical significance. The emergence of novel anti-CD20 monoclonal antibodies has sparked interest in their potential role in treating refractory MN. In a single-center study, obinutuzumab demonstrated high efficacy, with 94.4% of MN patients achieving partial or complete remission, even among those who previously responded poorly to rituximab [56]. The study observed a remission rate of 72.2% at six months and 88.9% at twelve months, surpassing the outcomes reported in similar studies [57]. Notably, obinutuzumab induced faster remission and showed promising results in refractory MN cases. Its good tolerability, even in patients who experienced severe reactions to rituximab, suggests lower immunogenicity. The enhanced efficacy of obinutuzumab may be attributed to its deeper B-cell depletion and distinct mechanisms of action, including greater direct cell death and improved antibody-dependent cytotoxicity compared to rituximab. However, further research is needed to confirm these findings and fully understand the underlying mechanisms. Future prospective studies and randomized controlled trials are necessary to validate these outcomes and explore the broader implications of obinutuzumab in MN therapy and epitope spreading, particularly in rituximab-resistant cases.

### 2.4. Evidence of Intermolecular Epitope Spreading in MN

Concerning intermolecular spreading, it is observed that most MN subtypes, distinguished by their specific antigens, tend to be mutually exclusive. This exclusivity raises intriguing questions about the mechanisms underlying autoimmune responses, which typically target a single antigen. The precise reasons for this selective targeting remain largely enigmatic. It is hypothesized that an individual’s class II HLA repertoire plays a role, with a lower probability of possessing multiple HLA risk alleles necessary for initiating and sustaining an autoimmune response to more than one antigen [32]. Occasional reports indicate dual antigen positivity, such as PLA2R and THSD7A, occurring in about 1% of cases [58], and considering the universal expression of PLA2R and THSD7A by human podocytes, the rarity of dual reactivity suggests that intermolecular epitope spreading is not solely driven by cell damage [32]. Other combinations include PLA2R with EXT1/2 [59], THSD7A with EXT1/2 [60], and NCAM1 with EXT1/2 [61]. However, there is no conclusive evidence that one autoimmune reaction precedes another, nor is there prognostic information available. This lack of definitive proof leaves the concept of intermolecular spreading speculative. Nonetheless, clinical cases of MN associated with autoimmune thyroiditis and Graves’ disease illustrate intermolecular spreading, where autoantibodies against both thyroid and kidney antigens suggest that autoimmunity in one organ can lead to the development of autoimmunity in another through epitope spreading [62]. Another notable discovery is that autoantibodies targeting intracellular antigens, especially anti-aENO autoantibodies, are most frequently observed in PLA2R1-positive patients with high anti-PLA2R1 titers or those identified as spreaders. This combined positivity is linked to a greater reduction in eGFR. The lower presence of anti-aENO autoantibodies in patients with low anti-PLA2R1 titers or non-spreaders indicates that autoantibodies against intracellular antigens likely develop as a secondary response following the initial formation of autoantibodies against PLA2R1 (or potentially THSD7A) [63].

### 2.5. Critical Reflections

As highlighted by Beck and Salant [64], while defining epitope spreading is conceptually straightforward, measuring specificities in a standardized and clinically meaningful way is technically challenging due to varying assay sensitivities. This variability likely explains the differing conclusions previously discussed. Factors such as construct design, expression systems, protein stability, denaturing conditions, and protein binding methods can alter epitope conformation and accessibility. Increased assay sensitivity can also detect irrelevant or cross-reactive epitopes, complicating the assessment of the humoral repertoire. The combined quantitative detection of specific IgG and IgG4 antibodies against PLA2R and its epitopes can identify epitope spreading, with specific IgG4 antibodies being more useful for prognosis [53]. Despite potential cross-reactivity, it is probable that multiple distinct PLA2R-ab specificities exist, reflecting distinct B cell pools that evolve over time. Longer disease durations can lead to epitope spreading, oligoclonal expansion, and affinity maturation, resulting in higher overall PLA2R-ab titers. High titers are associated with increased complement-mediated cytotoxicity [65], reduced treatment responsiveness, and worse clinical outcomes. Clinicians can use antibody titers as a practical guide for decision-making in the absence of epitope-specific immunoassays, serving as a proxy for epitope spreading [66,67]. Concurrently, research should focus on tracing the origins and evolution of the antibody repertoire, including epitope specificity. A deeper understanding of the evolution and variety involved in epitope spreading to any MN target antigen could enhance prognosis, clarify the initial triggering events in disease pathogenesis, and facilitate the development of antigen-targeted therapies.

## 3. Lupus Nephritis

Lupus nephritis (LN) represents one of the most severe manifestations of SLE, a chronic autoimmune disease characterized by widespread loss of immune tolerance to self-antigens featuring tissue damage caused by autoreactive T- and B-cells along with a vast repertoire of autoantibodies in genetically predisposed patients. The complexity of LN pathogenesis is increasingly understood to involve both intramolecular and intermolecular spreading. This process not only exacerbates autoimmunity but also contributes to the progression and severity of the diseases, significantly contributing to the heterogeneity and severity of renal involvement in lupus patients. In the early stages of lupus nephritis, immune complexes primarily form in the mesangium, corresponding to less severe forms of the disease, such as Class I or II lupus nephritis (ISN/RPS Classification [68]). However, as epitope spreading occurs, additional autoantibodies are produced that recognize and bind to new epitopes in different glomerular compartments. This leads to the deposition of immune complexes in the subendothelial and subepithelial spaces of the glomerulus, which are associated with more severe forms of lupus nephritis, such as Class III (focal) and Class IV (diffuse) lupus nephritis [69] (Figure 2).

### 3.1. Epitope Specificity in Lupus Nephritis

Traditionally, anti-dsDNA antibodies have been considered a hallmark of SLE; however, it has become clear that nucleosomes, the fundamental units of chromatin consisting of DNA wrapped around histone proteins, are the primary autoantigens driving the autoimmune response in LN [70]. It has been demonstrated that anti-dsDNA antibodies do not directly target glomerular structures such as laminin or type IV collagen [71]. Instead, these antibodies bind to nucleosomes that have been deposited in the GBM [72]. This interaction is critical because it enables the formation of immune complexes that contribute to the characteristic glomerular deposits seen in lupus nephritis. Proof that these nucleosome-mediated immune complexes are central to the pathology of LN is given through the demonstration that perfusion of nucleosome–autoantibody complexes into the kidneys of experimental animals led to significant glomerular binding and immune complex deposition, whereas purified, nucleosome-free antibodies did not exhibit the same binding affinity [73].

### 3.2. Evidence of Intermolecular Epitope Spreading in Lupus Nephritis

Extensive evidence from animal studies suggests that the breakdown of self-tolerance to key antigens, especially epitopes of small nuclear ribonucleoproteins (RNPs), such as Ro-RNP and snRNP, leads to epitope spreading and the onset of overt SLE [74,75,76,77,78] and LN [79]. Moreover, Singh et al. revealed, in lupus-prone mice, a sophisticated mechanism known as reciprocal T-B determinant spreading, where T cells spontaneously react to peptides derived from autoantibodies, specifically the VH regions of anti-DNA antibodies. These T cells, in turn, provide help to B cells that produce these antibodies, fueling a cycle of mutual activation. As T cells help B cells that share similar epitopes, the response is not limited to a single B cell population. Instead, it spreads to different B cells that may present variations of the original epitope, thus broadening the autoimmune attack and leading to the involvement of multiple autoantigens over time [80]. A study by Wu et al. investigated the role of the half-cryptic C1q-A08 epitope within the C1q protein involved in the classical complement pathway. Their finding showed that mice immunized with the C1q-A08 peptide not only developed anti-C1q-A08 antibodies but also showed evidence of epitope spreading, where the immune response extended to other parts of the C1q molecule, corroborating the complexity and severity of autoimmune responses in LN [81]. Moreover, it was demonstrated that histones H2B and H4 are not just passive participants but active drivers of LN through their interaction with pathogenic T-CD4+ cells. What is particularly striking is that these epitopes are strategically located in regions where histones bind to DNA, effectively shielding them from degradation during antigen processing. This protection allows these epitopes to be presented to Th cells in a highly immunogenic form, making them prime targets in the autoimmune cascade. Immunized mice with peptides from these critical regions of H2B and H4 not only triggered a stronger autoimmune response but also led to an earlier onset and more severe form of LN, suggesting that, once the immune system latches onto these protected, high-impact epitopes, the autoimmune response becomes much more aggressive and difficult to control [82]. Similar mechanisms are likely at play in human SLE and LN, as autoantibodies against a few specific autoantigens appear in the preclinical stage, with a significant expansion of autoantigenic targets at disease onset, as demonstrated by Arbuckle et al., who tested 130 patients for major SLE autoantibodies using indirect immunofluorescence and ELISA assays. They found that, on average, patients recognized 1.5 out of 7 autoantigens before diagnosis, increasing to 3 out of 7 at the time of clinical diagnosis, suggesting that epitope spreading occurs during the progression from preclinical to clinical SLE, contributing to the disease’s pathogenesis [10]. Nonetheless, longitudinal surveillance of the IgG autoantibody repertoire revealed that, contrary to the expected continuous expansion of autoantibody diversity due to epitope spreading, the overall autoantibody repertoire remained remarkably stable over time, challenging the notion of an unrestrained autoimmune amplification loop in established SLE [83]. However, the nuanced evolution in fine epitope specificity, particularly within antigenic complexes like U1-RNP, suggests that this intramolecular epitope spreading may still play a role in disease activity, especially preceding new organ involvement [83]. Furthermore, kidney tertiary lymphoid structures (TLSs), which are organized clusters of immune cells that form in non-lymphoid tissues in response to chronic inflammation or persistent immune activation, may play a significant role in defining epitope spreading in LN. Several studies highlighted that kidney TLS can promote a localized immune response against specific autoantigens that are overexpressed in the inflamed tissue [84]. For instance, antibodies produced by B cells within TLS were found to target vimentin [85], a protein that is overexpressed in the kidneys of affected individuals, and the tubulo-interstitium [86]. This suggests that TLS can facilitate the in situ, antigen-driven immune response, leading to the generation of new autoantibodies and contributing to the worsening of LN by enabling intermolecular spreading. The presence of TLS and the associated epitope spreading are linked with higher disease activity, chronicity indices, and poor response to treatment in LN patients, indicating that epitope spreading driven by TLS could be a marker of disease severity [84]. A clinical example of epitope spreading in LN was shown in a case report that described a rare presentation in a patient who was seronegative for typical lupus markers but positive for anti-glomerular basement membrane (anti-GBM) antibodies. Initially diagnosed with anti-GBM disease, the patient later developed combined focal proliferative and membranous lupus nephritis (Class III + V), as evidenced by sequential kidney biopsies. The first biopsy showed segmental and granular immune deposits, while the second revealed a full-house pattern with significant electron-dense deposits. The researchers hypothesize that the observed histopathological transition may be due to epitope spreading, potentially triggered by anti-GBM antibodies [87].

### 3.3. Therapeutic Implications

Several novel drugs studied for LN focus on targeting autoantigen presentation and the expansion of autoreactive lymphocyte clones rather than directly addressing inflammation. B-cell-targeted therapies, for instance, eliminate a key group of antigen-presenting cells, which play a significant role in immune responses. Tissue injury in autoimmune conditions involves a complex interplay of immune mechanisms, including immune complex formation, T-cell-mediated immunity, and various components of innate immunity, such as complement activation, Fc receptor signaling, and the actions of cytokines and chemokines at the peripheral tissue level. B-cell depletion therapies eliminate antigen-presenting cells and precursors to autoantibody-producing cells, indirectly reducing inflammation by preventing immune complex generation and intrarenal tertiary lymphoid tissue formation [88]. While these drugs may not show immediate effects in induction trials, they could be more effective in preventing future LN flares and minimizing kidney damage, suggesting their potential success in the maintenance of remission trials. Evidence supports the idea that drugs targeting autoimmunity, such as belimumab and abatacept, may be particularly suited for flare prevention rather than immediate renal response [89,90]. Belimumab, a human monoclonal antibody that inhibits B-cell-activating factor (BAFF or BLyS), was approved by the FDA in 2011 for SLE treatment due to its demonstrated efficacy in clinical trials [91]. A multinational phase 3 trial (BLISS-LN) involving 448 patients over 104 weeks evaluated the addition of belimumab to a triple immunosuppressive regimen, showing superior primary efficacy renal response rates compared to placebo, with similar adverse event rates [92]. No published evidence yet confirms that BLyS and APRIL therapeutic strategies directly inhibit B cell epitope spreading, aside from the lack of clinical signs and symptoms observed in trials. An ideal therapeutic approach for LN would combine anti-inflammatory and autoimmune-targeting therapies from the outset to achieve and maintain a long-term renal response while minimizing toxicity [88].

## 4. IgA Nephropathy

Immunoglobulin A nephropathy (IgAN) is the most common glomerulonephritis worldwide and is characterized by the deposition of aberrantly glycosylated IgA in the glomerular mesangium [93]. IgA nephropathy has been traditionally recognized as a systemic disease where the kidneys suffer damage as innocent bystanders [94], as evidenced by its frequent recurrence after transplantation. Interestingly, IgA glomerular deposits from a donor with subclinical IgA nephropathy have been observed to clear within weeks after the kidney is transplanted into a patient with a different kidney disease. This suggests that the underlying systemic factors driving IgA nephropathy may be key to its pathogenesis rather than the kidney itself being the primary site of disease origin [95].

### 4.1. The “Four-Hit Hypothesis”

Recent advancements in understanding the pathophysiology of IgAN have introduced the “four-hit hypothesis” [96]. This model begins with the production of galactose-deficient IgA1 (gd-IgA1) in Peyer’s patches and mesenteric lymph nodes (first hit). Nonetheless, high levels of gd-IgA1 alone are insufficient to cause glomerular damage. The disease requires the formation of IgG or IgA1 autoantibodies that recognize the terminal GalNac of gd-IgA1 as a neo-epitope [97] (second hit). These autoantibodies, primarily of the IgG isotype, possess an unusual sequence in the variable region of their heavy chains, likely due to somatic mutation [98], which enhances binding to the galactose-deficient glycans of gd-IgA1. The exact mechanisms leading to the formation of these anti-glycan antibodies remain unclear, but it is suggested that infections with bacteria expressing GalNac on their surface may trigger the production of glycan-specific antibodies through molecular mimicry, potentially explaining the frequent occurrence of hematuria alongside mucosal infections in IgAN patients [99]. It is worth noting that serum levels of anti-gd-IgA1 IgG correlate with proteinuria, gd-IgA1 levels, disease progression, and the risk of end-stage kidney disease (ESKD) [100,101,102]. These circulating immune complexes, composed of gd-IgA1, anti-gd-IgA1, and C3, then deposit in the glomerular mesangium (third hit) [103]. The deposition of immune complexes in the glomeruli triggers the activation of mesangial cells and the release of aldosterone, angiotensin II, pro-inflammatory cytokines such as interleukin-6, and growth factors like transforming growth factor-beta [96]. This cascade leads to mesangial cell proliferation and activation of the complement pathway and, ultimately, results in glomerular injury and interstitial fibrosis (fourth hit) [94].

### 4.2. Epitope Spreading in IgA Nephropathy: The Possible Connection

As of today, the concept of epitope spreading and antigen specificity of circulating IgA have not been emphasized in the pathogenesis of IgAN. Despite the proposed model, a fundamental question remains: why are immune complexes containing gd-IgA1 selectively deposited in the glomerular mesangial region? Considering their nature, these complexes would typically be expected to deposit randomly across various glomerular locations, similar to the patterns seen in lupus nephritis. Recent findings have shown that IgA binds to specific autoantigens expressed on mesangial cells, likely serving as an initial event in IgAN pathogenesis. This has led to a new conceptualization of IgAN as a tissue-specific autoimmune disease [104]. Additionally, it has been discovered that certain strains of oral commensal bacteria can induce the production of anti-mesangium IgA in normal mice when immunized with strong adjuvants, possibly through antigenic molecular mimicry. This raises important questions about whether the maintenance of circulating anti-mesangial IgA requires ongoing bacterial colonization and memory B cells reactive to these bacteria and whether immune tolerance is broken during the initial phase of IgAN. However, infection with bacteria that exhibit molecular mimicry does not always lead to autoantibody production, suggesting that the breakdown of immune tolerance, and possibly epitope spreading, is crucial for the development of autoantibodies [105,106]. The cytokine APRIL, which promotes B-cell survival and differentiation, may play a role in altering immune tolerance. Elevated APRIL levels correlate with disease severity in IgAN patients [107], and abnormal B cells expressing APRIL have been found in the tonsils of these patients [108]. The effectiveness of sibeprenlimab an anti-APRIL monoclonal antibody in treating IgAN further supports APRIL’s involvement in the disease’s pathogenesis [109]. Additionally, the B-cell activating factor belonging to the tumor necrosis factor family (BAFF), closely related to APRIL, is known to contribute to the breakdown of immune tolerance, suggesting that it may similarly disrupt tolerance and lead to the production of anti-mesangial IgA in IgAN through a response to antigens mimicking mesangial self-antigens [104]. Further research is needed to determine the role of epitope spreading and autoantigen targeting in human IgAN, which could open new avenues for definitive treatment.

## 5. ANCA Associated Vasculitis

Antineutrophil cytoplasmic autoantibody (ANCA)-associated vasculitis (AAV) are systemic vasculitis syndromes marked by inflammation and necrosis of small vessel walls. The etiology and pathogenesis of AAV are influenced by a combination of genetic, epigenetic, and environmental factors. Based on current clinical and experimental evidence, it is plausible that, in genetically predisposed individuals, various triggers can initiate the production of ANCA, which, within an inflammatory environment, can lead to tissue inflammation and vascular injury [110]. Typically, c-ANCA targets proteinase 3 (PR3) and is predominant in granulomatosis with polyangiitis (GPA), while p-ANCA targets myeloperoxidase (MPO) and is more common in microscopic polyangiitis (MPA), though these associations are not absolute [111]. Evidence suggests a pathogenic role for ANCA in systemic vasculitis, as patients who remain ANCA-positive during remission are at higher risk of relapse, and changes in ANCA titers often precede disease flares [112]. The variation in disease progression among patients with AAV may be influenced by the antibodies’ recognition of different binding sites, or epitopes, on their target antigens. These differences in binding specificity can affect the pathogenic potential of the antibodies. Consequently, variations in ANCA epitope specificity between patients or changes in epitope specificity within an individual patient over time may lead to differences in how the disease presents and progresses. Several functional characteristics of PR3– and MPO–ANCA have been identified, with changes in these characteristics occurring as the disease progresses, potentially indicating that ANCA begins to recognize different epitopes (Figure 3).

### 5.1. Evidence of Intramolecular Epitope Spreading in PR3–ANCA

PR3, a serine proteinase, is a highly folded protein maintained by four disulfide bridges, and it undergoes specific processing to become enzymatically active [113]. The characterization of PR3–ANCA interactions has been challenging due to the conformational nature of the epitopes that PR3–ANCA recognizes. Studies have shown that PR3–ANCA primarily binds to conformational epitopes, with some evidence suggesting the presence of linear epitopes as well [114]. However, the identified epitopes on PR3 recognized by PR3–ANCA, particularly in patients with GPA at initial disease presentation, seem to be limited to a few immunodominant regions, often located near the active site residues of PR3 [115]. This binding likely has functional consequences for PR3. Epitope mapping studies have been complicated by differences in PR3 processing and structure, but it is suggested that the epitopes recognized by PR3–ANCA might be cross-reactive with other antigen sources like microbial antigens or complementary protein fragments [116]. A novel and intriguing pathogenic mechanism has been proposed for AAV, suggesting that autoimmunity to PR3 might be initiated through an immune response against the antisense complementary peptide of PR3 [116]. Antibodies to this complementary PR3 could trigger an idiotype–anti-idiotype response, leading to the production of anti-idiotypic antibodies that cross-react with PR3, thereby functioning as autoantibodies. The functional characteristics of PR3–ANCA differ between inactive and active phases of the disease, indicating that the epitopes targeted by these autoantibodies can change over time [117]. In cross-inhibition studies, it was found that the epitopes recognized by PR3–ANCA at the time of diagnosis often differ from those recognized during later relapses [118]. Some patients experienced epitope spreading over time, while others showed a narrowing of targeted epitopes [119]. Additionally, PR3–ANCA that bind to both the proform and mature form of PR3 are more strongly correlated with disease activity in GPA than those that bind only to the mature form, further suggesting that epitope recognition by PR3–ANCA evolves as the disease progresses [120].

### 5.2. Evidence of Intramolecular Epitope Spreading in MPO–ANCA

Animal, in vitro, and clinical studies have confirmed the pathogenic role of MPO–ANCA in AAV, but the specific epitopes targeted by these antibodies remain incompletely understood. Although studies on MPO–ANCA interactions have been less extensive than those on PR3, it has been suggested that MPO–ANCA primarily recognize conformational epitopes, as their binding is resistant to mild denaturation but is destroyed by thermal denaturation [115]. Epitope mapping using recombinant deletion mutants of MPO has shown that most MPO–ANCA target up to three regions on the heavy chain of MPO, with none binding to the light chain [121]. Interestingly, patients with MPO–ANCA that recognizes only one or two epitope regions have a higher relapse rate than those recognizing all three regions, indicating a potentially worse prognosis [121]. Through recombinant MPO, an immunodominant epitope on the surface of MPO has been identified. However, the recognition of these epitopes may depend on the source of the antigen, as some MPO–ANCA sera fail to recognize recombinant MPO [122,123]. Another study identified seven humoral epitopes, predominantly located on the heavy chain of MPO, which are likely important in disease pathogenesis and manifestations [124]. Although it is known that MPO–ANCA primarily recognizes conformational epitopes, the role of linear epitopes in disease progression is less clear. A study conducted by Gou et al. investigating sera from AAV patients at different stages of the disease (initial onset, remission, and relapse) tried to map the specific epitopes recognized by MPO–ANCA [125]. More than half of the sera from patients contained one or more linear epitopes, while others likely contained only conformational structures. Notably, the light chain of MPO, typically hidden within the MPO dimer structure, was recognized in a significant portion of patients with more severe renal dysfunction and systemic disease, suggesting that, when the light chain is targeted, it may indicate more extensive epitope spreading and more severe disease. Furthermore, the study found that epitope recognition remained consistent between initial onset and relapse, suggesting a role for immunological memory in disease recurrence. However, the number of recognized epitopes tended to decrease during remission. Another study further corroborated the finding that the pathogenicity of MPO–ANCA in vasculitis is determined by epitope specificity, explaining why natural MPO autoantibodies exist in healthy individuals and why ANCA titers do not always correlate with disease activity [126]. A specific linear epitope on MPO was identified that is targeted by autoantibodies in ANCA-negative patients and often masked by serum proteins, providing a new understanding of the ANCA-negative subset of vasculitis. These findings highlight the role of epitope spreading in disease progression and suggest how initially asymptomatic autoantibodies evolve to contribute to active disease as new, pathogenic epitopes are unmasked and recognized by the immune system.

### 5.3. Critical Reflections

PR3–ANCA and MPO–ANCA recognize a limited set of epitopes on their respective targets, showing the role of intramolecular spreading in AAV. Epitopes recognized by ANCA may change over the course of the disease. Identifying and characterizing these relapse- or disease-inducing epitopes are crucial for understanding the initiation and reactivation of AAV. Discovering such epitopes could pave the way for the development of epitope-specific therapeutic strategies. Additionally, it could help identify the foreign-like epitope responsible for triggering the disease in susceptible individuals.

## 6. Anti Glomerular Basement Membrane Disease

Anti-glomerular basement membrane (anti-GBM) disease or Goodpasture syndrome is a small vessel vasculitis affecting glomerular capillaries, pulmonary capillaries, or both, with deposition of anti-GBM autoantibodies along the GBM [127]. This rare disease has an incidence of 1.5 per million and prevails in the male sex [128]. The most common presentation is rapidly progressive glomerulonephritis with a doubling of serum creatinine, sometimes within a few days [129], accompanied by hematuria and from mild to nephrotic proteinuria [130]. Often, for several weeks before diagnosis, patients experience nonspecific symptoms such as malaise, fatigue, weight loss, and fever [131]. Pulmonary hemorrhage is also common, especially in smokers, and can lead to respiratory failure [132]. There are several variants of the disease, and caution is needed to make a correct diagnosis. Such variants include overlap with vasculitis associated with anti-neutrophil cytoplasmic antibody and membranous nephropathy, as well as anti-GBM occurring de novo after renal transplantation.

### 6.1. Evidence of Epitope Specificity and Spreading in Anti-GBM Disease

Diagnosis of anti-GBM disease is based on histology and detection of anti-GBM antibodies, either via direct immunofluorescence (IF) or solid-phase methods. All patients with anti-GBM disease have autoantibodies that target two distinct epitopes on the a3 chain of type IV collagen: residues 17–31, called EA, and residues 127–141, called EB [133,134,135,136]. Jun-liang Chen et al. linked EA and EB reactions, with EB as an independent risk factor for renal failure. Intramolecular epitope spreading could occur before disease onset [137]. Lanlin Chen et al. showed a T-cell epitope of 3(IV) NC1 induced experimental autoimmune glomerulonephritis capable of spreading to the 4(IV) NC1 domain with minimal or no reactivity toward other collagen chains or glomerular constituents [138]. Healthy individuals have low-affinity antibodies for the same epitopes, but Tregs typically prevent the development of pathogenic high-affinity autoantibodies. Peptides derived from the α 3 chain presented by the HLA-DR15 antigen do not have the ability to promote the development of such Tregs [139]. Pathogenic anti-GBM autoantibodies target the noncollagen (NC1) domains of collagen α3α4α5 (IV), a major GBM component. Conformational epitopes are sequestered within the hexamer complex α3α4α5NC1 formed by collagen IV chains. Autoantibodies selectively bind and dissociate monomeric subunits, while dimeric NC1 subunits resist dissociation. This suggests that structural changes occur in the α3α4α5NC1 hexamer unmask epitopes, disrupting tolerance and triggering autoimmunity [140]. Numerous studies have identified B-cell epitopes recognized by anti-GBM antibodies, including the 36-amino-acid sequence of noncollagenous domain 1 (NC1) of type IV collagen chain 3 (Col4 3NC1), known as the Goodpasture antigen [141]. Animal models have shown that immunization with Col4 3NC1 can induce an immune response [142,143,144], though W. Kline Bolton et al. found that this alone is insufficient to induce GN [145]. In addition to Col4 3NC1, several GBM proteins, including several type IV collagen chains, collagen domains, and the S7 domain of type IV collagens and other noncollagen components of GBM, have been identified as Ag recognized by autoantibodies from patients with GBM [146,147,148,149,150]. Studies have shown that anti-GBM antibodies primarily react with the three-dimensional (3-D) conformation of native antibodies due to the complex quaternary structure of GBM [151,152].

The role of T cells in glomerular damage became clearer when Jean Wu et al. demonstrated that the T-cell epitope pCol28–40 induces severe GN and triggers an autoantibody response through B-cell epitope spreading. This suggests that anti-GBM antibodies may arise from T-cell-mediated glomerular damage [153]. Arends et al. identified three critical amino acids in this nephritogenic epitope, suggesting molecular mimicry with microbial peptides [154]. Shui-yi Hu et al. identified a nephritogenic T-cell epitope, P14 (a3127-148), which induced anti-GBM nephritis and epitope spreading in WKY rats [155]. Robertson et al. also identified pCol(28–40) as a nephritogenic T-cell epitope, proposing that T-cell-mediated glomerular injury may trigger the activation of GBM-specific B cells [156]. Shui-yi Hu et al. later identified five epitopes: α3(IV)NC1127-148 (P14), α3(IV)NC1159-178, α3(IV)NC1179-198, α3(IV)NC1189-208 (P19), and α3(IV)NC1141-154. P14 and P19 were more present in patients than in healthy controls. They highlighted that P14 was a reciprocal epitope of T and B cells, implying its initial role in the epitope spreading process [157]. Based on all these considerations, Yue Shi et al. hypothesized that the design of a modified peptide, starting from P14 (a3127-148) and replacing critical pathogenic residues with non-pathogenic ones (based on homologous regions in the a1NC1 chain of type IV collagen, known to be non-pathogenic), could provide a therapeutic option for anti-GBM GN. This approach confirmed the feasibility of modulating T-cell activation for the treatment of Goodpasture’s disease and could provide new insights into the treatment of autoimmune kidney disease in the future [158].

### 6.2. Clinical Evidence of Intermolecular Spreading in Anti-GBM Disease

Around 20–30% of patients with anti-GBM disease have coexisting anti-MPO autoantibodies, which are often associated with disease severity. The potential common pathogenic mechanism between anti-glomerular basement membrane antibodies and anti-MPO autoantibodies in anti-glomerular basement membrane disease is still not well understood; however, Jian-nan Li et al. showed that many patients with anti-GBM disease have autoantibodies against MPO279-410 linear peptides, suggesting that “double positivity” is not coincidental. This indicates that autoreactive T cells against MPO might activate B cells, driving an immune response to linear MPO peptides [159]. ANCA-associated vasculitis can also complicate anti-GBM disease. Yuka Nishibata et al. suggested that ANCA-activated neutrophils release proteases that digest Col (IV), exposing α3(IV)NC1 and leading to anti-GBM antibody production [160]. A rare entity involves lupus membranous nephritis associated with anti-GBM antibodies in patients serologically negative for ANA and anti-dsDNA, with normal complement levels. Class V lupus nephritis leads to complement activation and GBM thickening, resulting in nephrotic-range proteinuria. Zhang et al., using a mouse model of membranous glomerulopathy, demonstrated that mice immunized with rh-α3NC1 developed proteinuria, GBM thickening, and immune deposits, showing that GBM antibodies α3(IV) are involved in membranous glomerulopathy [161].

### 6.3. Critical Reflections

In conclusion, the phenomenon of epitope spreading in anti-GBM disease plays a critical role in its pathogenesis, highlighting the dynamic nature of the autoimmune response. As T-cell-mediated damage to the glomerular basement membrane occurs, it can expose new epitopes, leading to a cascade of immune responses that exacerbate the disease. Understanding this process of epitope spreading offers promising therapeutic implications, as it suggests that early intervention to halt the spread of immune responses to additional epitopes could prevent the progression of the disease. Targeted therapies that inhibit this spreading could potentially preserve renal function and improve outcomes for patients, paving the way for more precise and effective treatments in autoimmune kidney diseases.

## 7. Conclusions

The phenomenon of epitope spreading stands as a cornerstone in the pathogenesis of autoimmune glomerulonephritis, illustrating the transition from a targeted immune response to a more expansive, destructive process (Table 1). The detailed examination of intramolecular and intermolecular epitope spreading across various autoimmune conditions underscores the complexity and adaptability of the immune system’s response to tissue damage. As demonstrated in diseases like membranous nephropathy, lupus nephritis, and ANCA-associated vasculitis, the progression from initial antigenic targets to secondary, structurally unrelated epitopes not only amplifies the autoimmune response but also complicates clinical outcomes and therapeutic approaches. The role of epitope spreading in exacerbating autoimmune glomerulonephritis highlights the importance of early and targeted intervention. By intercepting the immune response before it broadens to include additional epitopes, there is the potential to significantly mitigate tissue damage and disease progression. This insight has profound therapeutic implications, suggesting that the development of treatments capable of modulating the immune system at these critical junctures could lead to more effective management of the disease. Moreover, the clinical relevance of epitope spreading extends beyond immediate therapeutic strategies. It serves as a potential marker for disease prognosis, offering a window into the likely trajectory of disease severity and responsiveness to treatment. As research continues to explore the intricacies of epitope spreading, the potential to refine diagnostic and therapeutic tools becomes increasingly clear, paving the way for precision medicine approaches that could transform patient outcomes. In essence, understanding epitope spreading not only illuminates the pathophysiology of autoimmune glomerulonephritis but also sets the stage for innovative interventions that could prevent or halt the disease’s progression. The ongoing research into this complex phenomenon promises to unlock new avenues for treatment, offering hope for more personalized and effective management strategies in the battle against autoimmune diseases.

## 8. Materials and Methods (Concise)

### 8.1. Inclusion and Exclusion Criteria

This narrative review implemented stringent inclusion and exclusion criteria to ensure the selection of high-quality and relevant studies. The inclusion criteria comprised original clinical and preclinical studies focused on epitope spreading in autoimmune glomerulonephritis. Additionally, review articles were included to provide comprehensive background and context. Only articles published in English were considered. Exclusion criteria were meticulously designed to filter out studies that were irrelevant or of lower quality. Non-human studies were also excluded unless they provided significant insights into the biological mechanisms relevant to human disease. Articles not available in full text were omitted to ensure a thorough evaluation of all included studies.

### 8.2. Search Strategy

A comprehensive search strategy was developed to identify pertinent studies across several databases, including PubMed, Scopus, Web of Science, and the Cochrane Library. Google Scholar was also utilized to capture supplementary sources and grey literature, thereby expanding the scope of the search. Tailored search strings were constructed for each database to maximize the retrieval of relevant studies.

### 8.3. Data Extraction

Data extraction was performed using a standardized form designed to collect all relevant information from each included study. This form captured details such as authorship, year of publication, study type, methodologies, and key results.

### 8.4. Data Synthesis

Data synthesis was approached narratively, integrating qualitative data to elucidate the mechanisms of epitope spreading, clinical evidence, and therapeutic implications in autoimmune glomerulonephritis. This narrative approach allowed for a comprehensive and cohesive presentation of findings, highlighting key insights and trends across the included studies.

## Figures and Tables

**Figure 1 ijms-25-11096-f001:**
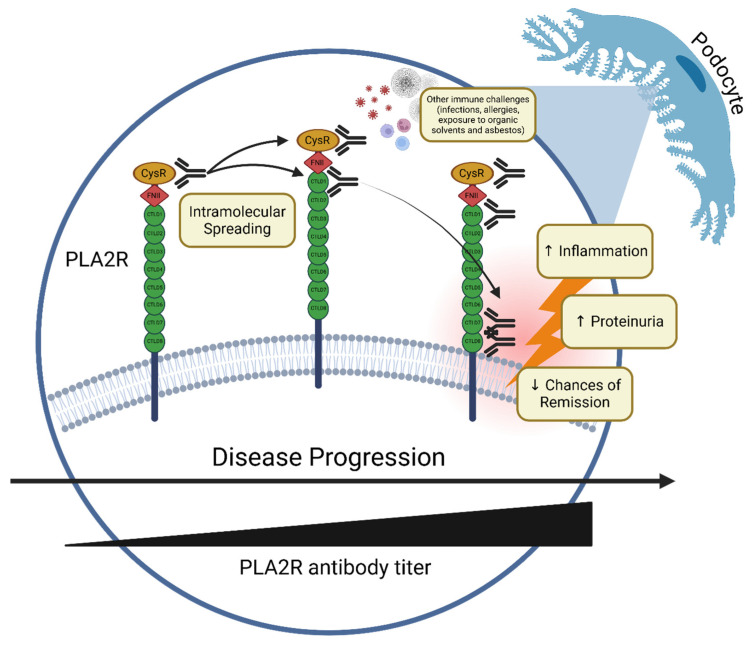
**Intramolecular epitope spreading in PLA2R-associated membranous nephropathy.** The progression of intramolecular epitope spreading in PLA2R-associated membranous nephropathy (MN). Initially, autoantibodies target the CysR domain of the PLA2R receptor on podocytes. As the disease advances, antibody recognition extends to additional domains, including CTLD1, CTLD7, and CTLD8, correlating with an increase in antibody titers and disease severity. This spreading drives podocyte injury, foot process effacement, and disruption of the glomerular filtration barrier, resulting in proteinuria. External immune triggers may accelerate this process, worsening clinical outcomes and reducing the likelihood of remission. (Created in BioRender.com).

**Figure 2 ijms-25-11096-f002:**
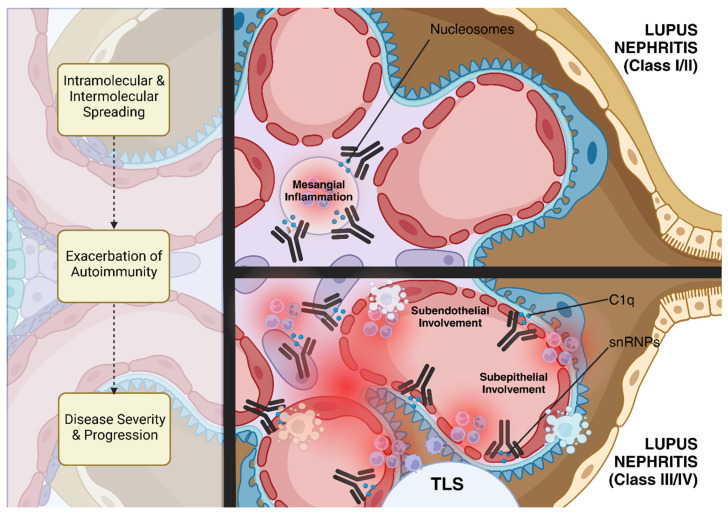
**Epitope spreading in lupus nephritis.** This diagram illustrates the process of epitope spreading in lupus nephritis. Early in the disease, nucleosome–autoantibody complexes form in the mesangium, causing mild kidney damage. As the disease progresses, epitope spreading occurs, leading to the production of additional autoantibodies targeting histone proteins and other glomerular autoantigens, such as snRNP and C1q. This results in immune complex deposition in the subendothelial and subepithelial spaces, contributing to more severe forms of lupus nephritis (Class III/IV). This process not only exacerbates autoimmunity but also contributes to the progression and severity of the diseases, significantly contributing to the heterogeneity and severity of renal involvement in lupus patients. Furthermore, kidney tertiary lymphoid structures (TLSs) can promote a localized immune response against specific autoantigens overexpressed in the inflamed tissue, in association with epitope spreading, higher disease activity, and poor treatment response. (Created in BioRender.com).

**Figure 3 ijms-25-11096-f003:**
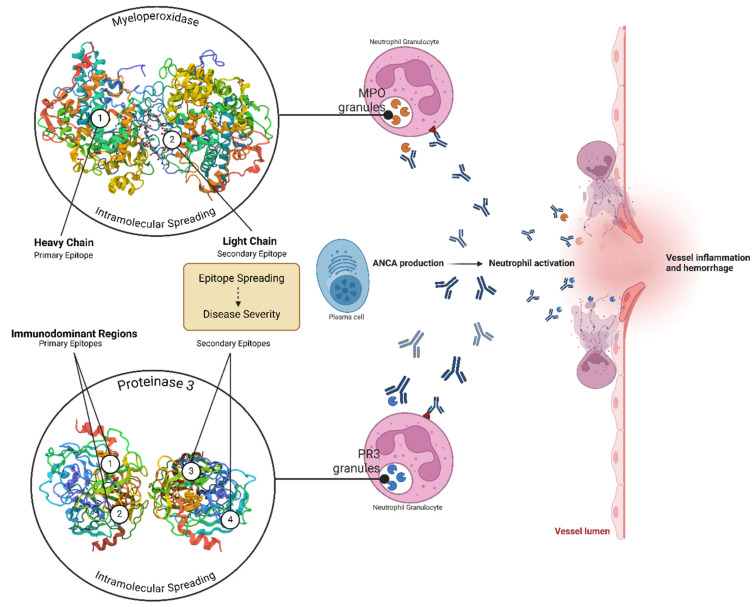
**Epitope spreading in ANCA-associated vasculitis.** The image illustrates the process of epitope spreading in ANCA-associated vasculitis (AAV). Initially, PR3–ANCA and MPO–ANCA antibodies target specific epitopes on PR3 and MPO, respectively. As the disease progresses, intramolecular epitope spreading occurs, leading to the recognition of new epitopes. This spreading contributes to increased vascular injury and inflammation, playing a crucial role in disease progression and relapse. (Created in BioRender.com).

**Table 1 ijms-25-11096-t001:** Epitope spreading in immune-mediated glomerulonephritis: mechanisms, clinical Implications, and therapeutic approaches.

Disease	Epitope Spreading	Molecular Targets	Clinical Implications
Membranous Nephropathy	Intramolecular, intermolecular (speculative)	PLA2R (CysR, CTLD1–CTLD8).	Worse prognosis, greater proteinuria, and higher risk of progression to ESRD.
Lupus Nephritis	Intramolecular and intermolecular	Nucleosomes, histones, snRNPs	Worsens renal involvement
IgA Nephropathy	Intermolecular (speculative)	Mesangial IgA1, external antigens	Limited evidence, potential role in immune complex formation
ANCA-Associated Vasculitis	Intramolecular	PR3, MPO	Higher relapse rate, severity of vascular injury
Anti-GBM Disease	Intramolecular	α3(IV)NC1, other α(IV) chains	Faster kidney failure progression

Overview of epitope spreading in autoimmune glomerulonephritis, summarizing the types, main targets, spreading mechanisms, and clinical impacts for five key pathologies.

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
