# Peer review of "Epitope Spreading in Immune-Mediated Glomerulonephritis: The Expanding Target"

_ijms, 2024, doi:10.3390/ijms252011096_

Round 1
Reviewer 1 Report
Comments and Suggestions for Authors
Dear authors:
This article entitled Epitope Spreading in Immune-Mediated Glomerulonephritis: The Expanding Target addresses a current and relevant topic for nephrology. The authors present the various topics in an orderly manner, supported by adequate and modern iconography. An adequate bibliographic search was carried out that supports the various topics discussed. Each pathology is adequately addressed.
Observations: In the revised document, corrections made by the authors remained, particularly in some titles.

Reviewer 2 Report
Comments and Suggestions for Authors
Strizzi C.T. et al. have conducted a high-quality and insightful review on immune-mediated glomerulonephritis. They effectively explained the key mechanism behind the progression of autoimmune glomerulonephritis, particularly the role of epitope spreading, which could serve as a potential therapeutic target in autoimmune glomerular diseases.
The review is well-structured and clearly written, with figures that are easy to understand and enhance the overall reading experience. I would only recommend adding a table to compare the different pathologies, such as lupus nephritis, anti-glomerular basement membrane disease, ANCA vasculitis, and IgA nephropathy. Additionally, I suggest changing the section titles from all capitals (e.g., "3. LUPUS NEPHRITIS") to title case (e.g., "3. Lupus nephritis") for consistency and improved readability. Finally, I recommend mentioning the use of the BioRender program in the figure legends to acknowledge the tool used for creating the illustrations.
Author Response
Dear Reviewer, Thank you for your thoughtful and constructive feedback on our manuscript. We are grateful for your positive evaluation of our work and your insightful suggestions to enhance the manuscript's clarity and comprehensiveness.- Addition of a comparative table: We have included a detailed table that compares the different autoimmune glomerular diseases. This table highlights the key differences and commonalities in their pathophysiology, with a particular focus on the role of epitope spreading, as per your suggestion. We believe this addition will further enrich the reader’s understanding of these complex conditions.
- Title case for section headings: We have revised all section titles to title case to ensure consistency and improved readability throughout the manuscript.
- Acknowledgment of BioRender: As recommended, we have added acknowledgments in the figure legends, specifying that the illustrations were created using BioRender.
Reviewer 3 Report
Comments and Suggestions for Authors
Strizzi et al delivered a narrative review on epitope spreading in several autoimmune mediated kidney diseases including lupus nephritis and ANCA vasculitis. Their hypothesis is interesting and the authors provide a large number of articles that support this possibility. Although the review is quite long, it is very complete and it has been a while that I've read such a good review. I therefore have no comments and believe the manuscript can be published as is. I look forward to further evidence to support their hypothesis in the future.
Reviewer 4 Report
Comments and Suggestions for Authors
Line 262: ‘NIAT’ does not have to be abbreviated since it appears only once in the main document.
Line 270: ‘UPCR’ should be spelt out on its first mention.
Line 474: it is not clear what ‘these drugs’ exactly means in the context: all drugs included in B-cell depletion regimen, or all drugs administered for lupus nephritis?
Line 517: ‘interleukin’ does not have to be abbreviated since it appears only once in the main document.
Line 518: ‘transforming growth factor’ does not have to be abbreviated since it appears only once in the main document.
Line 773: I think this paper is narrative review rather than systematic review.
Author Response
Dear Reviewer,
Thank you for your valuable feedback on our manuscript. We appreciate your careful review and have made the necessary revisions to address each of your points:- Line 262 – Abbreviation of ‘NIAT’: We have removed the abbreviation for "NIAT" as it appears only once in the manuscript, in line with your suggestion.
- Line 270 – ‘UPCR’ Spelling Out: "Urine Protein-to-Creatinine Ratio (UPCR)" is now spelled out upon its first mention, providing clarity for the readers.
- Line 474 – Clarification of ‘These Drugs’: We have revised the text to clarify the context, specifying whether it refers to all drugs included in the B-cell depletion regimen or all drugs used for lupus nephritis. The sentence now clearly references the B-cell depletion regimen.
- Line 517 – Full term for ‘Interleukin’: We have spelled out "interleukin" in full, as it appears only once in the manuscript.
- Line 518 – Full term for ‘Transforming Growth Factor’: "Transforming growth factor" is now spelled out in full, following your suggestion.
- Line 773 – Narrative Review Clarification: We agree with your observation that this paper is more appropriately classified as a narrative review rather than a systematic review. We have amended the text accordingly.